# Decentralized Clinical Trials: Governance, Ethics and Medico-Legal Issues for the New Paradigm of Research with a Focus on Cardiovascular Field

**DOI:** 10.3390/medsci13040222

**Published:** 2025-10-07

**Authors:** Elena Tenti, Giuseppe Basile, Claudia Giorgetti, Diego Sangiorgi, Elisa Mikus, Gaia Sebastiani, Vittorio Bolcato, Livio Pietro Tronconi, Elena Tremoli

**Affiliations:** 1Cardiovascular Department, Maria Cecilia Hospital, GVM Care & Research, 48033 Cotignola, Italy; 2Department of Biomedical Sciences and Public Health, University “Politecnica delle Marche” of Ancona, 60124 Ancona, Italy; 3CIRSFID, Department of Legal Studies, University of Bologna, 40126 Bologna, Italy; 4Astolfi Associati, Law Firm, 20122 Milano, Italy; 5Maria Beatrice Hospital, GVM Care & Research, 50121 Florence, Italy; 6Department of Health and Life Sciences, European University of Rome, 00163 Rome, Italy

**Keywords:** informed consent, decentralized clinical trials, digital technology, health literacy

## Abstract

The evolution of decentralized clinical trials, driven by advanced digital technologies, is transforming traditional clinical research. It introduces innovative methods for informed consent, remote patient monitoring, and data analysis, enhancing study efficiency, validity, and participation while reducing patient burden. Some clinical procedures can be conducted remotely, increasing trial accessibility and reducing population selection biases, particularly for cardiovascular patients. However, this also presents complex regulatory and ethical challenges. The article explores how digital platforms and emerging technologies like block chain, AI, and advanced cryptography can promote traceability, security, and transparency throughout the trial process, ensuring participant identification and documentation of each procedural step. Clear, legally compliant informed consent, often managed through electronic systems, both for research participation and data management in line with GDPR, is essential. Ethical considerations include ensuring participants understand trial information, with adaptations such as simplified language, visual aids, and multilingual support. The transnational nature of decentralized trials highlights the need for coordinated regulatory standards to overcome jurisdictional barriers and reinforce accountability. This framework promotes trust, shared responsibility, and the protection of participants rights while upholding high ethical standards in scientific research.

## 1. Introduction

The most recent definition of Decentralized Clinical Trials (DCTs) comes from the FDA (Food Drug Administration, US Medicines Agency) and MHRA (Medicines and Healthcare Products Regulatory Agency, UK Medicines Agency). DCTs are clinical studies that “through the use of telemedicine, digital health tools, and other information technology devices and tools, carry out some or all clinical procedures in areas distant from the practice location and integrate the health ecosystems and medical equipment already available for patient (s) to facilitate its participation to the trial itself” [1]. This broad definition reflects the progress and integration of digital technologies in healthcare. The COVID-19 pandemic has significantly and further influenced clinical research, boosted digitalization, and led to adaptations in ongoing and new trials through the decentralization of key elements, including remote approaches for consent (Figure 1) [2,3].

In general, as specified by European regulations on the applications of telemedicine and e-Care services, and further implemented by each Country, the main purpose of remote and decentralized clinical trials is to place the participants at the center of the trial, facilitating its participation while reducing the struggles and associated ultra-costs.

The decentralization of the clinical trial is aimed at achieving the following items:to increase the number of participants in each clinical trial and to collect useful data from participants in less time.to minimize the inconveniences and health risks for participants (and their caregivers) due to participation in the clinical trial by avoiding frequent travel and hospitalization.to improve participant adherence to trial procedures (compliance) and adverse events monitoring (safety).to easily share the data obtained from the clinical research studies using interactive platforms and analytics systems compliant by design and by default.

From this perspective, the purpose of digital technology applied to DCTs, is to promote participation, facilitate communication, and reduce practical difficulties, rather than focusing solely on AI. Specific applications include:
recruitment, administration of informed consent both for privacy disclosure and health treatment consenting, and further participation in the study take place totally or partially (hybrid trial) remotely.the collection of data—clinical, biological, environmental, self-reported (outcomes, i.e., through participant self-assessment)—is implemented using telemonitoring, tele visits, tele assistance, and ePROs (electronic patient-reported outcome).the wide possibility of analysis of the data, as well as easily shareable in RWD (Real World Data) [4].

DCTs face several ethical, medico-legal, and practical challenges related to the process of information and consenting. This short review discusses these aspects with a particular focus on the cardiovascular field.

## 2. Discussion

### 2.1. Normative Principles and Ethical Requirements

Ensuring compliance with EU and national laws, regulations, and established standards for clinical trials is essential, together with the integration of requirements for the decentralization of clinical trials. Key frameworks to adhere with are the clinical trial regulation and guidelines, as the International Council for Harmonization of Technical Requirements for Registration of Pharmaceuticals for Human Use (ICH) E6 on Good Clinical practice (2025) and E8 General considerations for clinical studies (2022), applicable Good Manufacturing Practice (GMP) provisions, and Good Distribution Practice (GDP) principles. Additionally, adherence to ethical and scientific principles, such as those outlined in the updated Declaration of Helsinki and WMA declaration of Taipei on ethical considerations regarding health databases and biobanks 2016, is essential.

This is relevant also for medical devices, including in vitro diagnostics (IVDs), frequently used in clinical trials: their use must comply with the applicable medical device legislation, such as the Medical Device Regulation (MDR) EU no. 2017/745, the In Vitro Diagnostic Directive 98/79/EC, and/or the In Vitro Diagnostic Regulation (IVDR) EU no 2017/746 [5]. Compliance with single State regulation on off-label drugs must also be considered. Off-label use is, however, recognized as a concept by EU pharmaceutical law (recital 2 of Pediatric Regulation and pharmacovigilance provisions in Directive 2010/84/EU) [6]. When biological samples too are collected, a part of the issues related to overall management to be linked with bio banking systems and regulations [7].

Decentralized trials introduce additional regulatory complexities. Key challenges include managing data across multiple jurisdictions, ensuring remote consent procedures meet legal and ethical standards, and maintaining consistent oversight when participants interact with study personnel virtually rather than in-person. Addressing these challenges requires careful planning and the adoption of robust systems to ensure both compliance and participant protection.

### 2.2. European Recommendations for DCTs

While several European countries have issued guidelines or regulatory frameworks for decentralized clinical trials, some countries such as China, Taiwan, Hong Kong, South Africa, Latin America still lack specific regulations, creating variability in implementation and oversight across the continent.

The European Commission (EC), along with the European Medicines Agency (EMA), released in 2022 the “Recommendation paper on decentralized elements in clinical trials”, as part of the Accelerating Clinical Trials in the EU (ACT EU) and alongside the full applicability of Regulation (EU) 2014/536 and the Clinical Trial Information System. These recommendations are aimed at providing Member States with more detailed guidance on implementing procedures for conducting clinical research activities outside traditional clinical trial centers [8]. In fact, except for temporary measures introduced during the COVID-19 pandemic to facilitate study continuation, many European countries have not yet implemented specific regulations on decentralized clinical trials. While not formally binding, they serve for standardization and guidance. The overall EU initiative is aimed at supporting the implementation of new testing regulations and bolstering the modernization and development of the European testing ecosystem, particularly focusing on trials involving multiple countries and employing digital technologies. Numerous decentralized features have been integrated into clinical trials involving medicinal products, electronic registries, wearables, telecommunication, and virtual appointments. The utilization of these decentralized elements varies significantly based on the trial’s nature, participant demographics, the targeted illness, participant health status, medical product type and characteristics, and developmental stage. Therefore, it is crucial to evaluate and incorporate these elements individually and collectively when strategizing and executing decentralized approaches.

### 2.3. Participant Information Processing

Emerging standards regarding the most appropriate ways to use novel approaches for approval (i.e., interaction design and choice architecture approaches) allow electronic consent to become a preferred choice to fulfil the scope of informed consent [9]. Electronic consent can be relevant not only to acquire informed consent for research, but also to obtain a legal basis for data processing under EU privacy norms (Figure 2).

Special attention should, in fact, be given to compliance with the General Data Protection Regulation (GDPR EU no 2016/679) to safeguard participant data privacy and security throughout the trial process. In this perspective, the precise identification of data processors and data controllers (the latter usually consists of the sponsor/clinic-institution of the investigator) is necessary as a first security guarantee, whereas in the second instance, as already in use in ordinary clinical trials, the monitoring of authorized participants and the definition of access privileges are needed [10]. Where necessary, the appointment of a data protection officer should also be promoted. This is more so in the hypothesis of enrolment and sharing across Europe. Platform privacy compliance must then be supported, and research purposes must be well-defined and projected so as to grant dataset effective use and sharing options. It is evident that non-profit organizations, companies, scientific societies, and professionals within the research context are highly prone to dissemination and collaboration, further increased by the DCT itself, giving rise to an augmented risk of data breach [11]. Legal complexity here necessitates the expertise of specialized professionals for comprehensive risk analysis, the effective structuring of appropriate procedures, effective document strategies, and the procurement of essential technical and organizational resources.

### 2.4. Consenting for Health Research Participation

The informed consent discussed here refers specifically to participation in research and is distinct from consent to clinical treatment. Informed consent is essential in health care practice, and even more so in research, where the choice depends more on the subject’s autonomy than on a purely clinical decision based on personal health needs.

Therefore, in addition to clinical relevance, comprehensive and detailed information about the purpose and scope of the research and the collective interest at stake is required. In European Union Regulation (EU) no 536/2014, informed consent is defined as “the free and voluntary expression of a participant of their willingness to participate in a specific clinical trial, after being informed of all aspects of the clinical trial relevant to the participant’s decision to participate or, in the case of minors and incapable individuals, the authorization or agreement of both their respective legally designated representatives to include them in the clinical trial” [12,13]. It is necessary to provide the participant with correct and understandable information to share treatment choices within the trial purpose [14]. The information requires considering times, places, and appropriate language methods. Also, information should be provided to all the participants, since it is important to consider that the time spent during the communication between doctor and participant is part of the process and empowers the participant themselves [15,16]. This is even more so given globalization and the participatory and relational nature of consent in some cultures.

In this context, it is relevant to consider participants that are socially conditioned to accept also legal provisions without reading them. By using the modern internet, mobile phones, and other computerized resources, the participants frequently click “agree” in only a short time. Thus, to prevent electronic informed consent from being ineffective or irrelevant, the information provided must be clear, understandable, simple but exhaustive. Interactive manners could also be sought [17]. Appropriately designed electronic tools for information can promote understanding and meaningful autonomous decision-making, despite the lack of physical interaction between study personnel and participants. It should also be traceable, and in this case, video recorded and signed through electronic consenting, incorporating means to verify the identity of the person giving consent (to assess that is the entitled participant). Minors’ involvement in the information pathway and decision-making must be sought, even if legal representatives are empowered for consenting [18,19]. According to EMA guidelines, the information provided to the potential participant in the study should clearly state that the study includes research. In addition, in the document the following items should be discussed: the purpose of the study, the study treatment(s) and the likelihood of random assignment to each treatment, the chances of success or possible existing clinical alternatives, the procedures that must be followed for the study, including all invasive ones, the participant’s responsibilities, reasonably foreseeable risks or complications/adverse events to the participant and, if applicable, for the embryo, fetus, or infant, reasonably foreseeable benefits [20]. Participants in fact show a lack of or a reduced understanding of research form [21], particularly regarding benefits/risks of surgery and anesthesia, and alternative treatment options [22,23]. Last, participants must know that their participation in the study is voluntary and that they can refuse to participate or withdraw from it at any time without any penalty or loss of benefits to which they would otherwise be entitled. Beyond procedural compliance, it is essential to actively promote genuine participant autonomy in research. This involves creating opportunities for meaningful dialogue, addressing questions and clarifying potential misunderstandings, and ensuring participants’ concerns are fully considered. Researchers must be aware of the risks of miscommunication, especially in vulnerable populations, and foster an environment that empowers participants to make informed decisions that truly reflect their values and preferences. The ethical robustness of decentralized trial processes depends not only on providing information, but also on ensuring understanding, engagement, and respect for participants’ autonomous decision-making.

Moreover, personalizing the informed consent process—adapting communication methods, timing, and formats to each participant’s needs—enhances ethical quality and supports genuine autonomy, rather than serving solely logistical purposes.

### 2.5. Health Literacy: A Matter of Ethics and Equity

Health literacy is a fundamental element in participants’ decision-making. The ethical dimension of health literacy is underscored by its intimate connection to the principle of informed consent [24]. Ensuring that participants fully understand the implications of a medical procedure or discover a new marker of pathology is not just a matter of good communication, but of respecting individuals’ self-determination rights. It now encompasses a range of cognitive and social skills that allow individuals to access, understand, and act on health information.

It is important to distinguish health literacy, the cognitive and social skills enabling individuals to access, understand, and act on health information, from health awareness, which refers more broadly to general knowledge or consciousness about health issues. While awareness may raise attention to health topics, literacy is what allows participants to make informed, autonomous decisions. Integrating both concepts in DCTs ensures not only that participants are aware of trial participation, but also truly understand and engage with the information provided.

The rise of digital technologies, such as DCTs, presents both opportunities and challenges for health literacy (Table 1).

These trials often rely on remote communication methods and digital platforms to collect informed consent. While they can make participation more accessible, they also raise concerns about ensuring that participants truly understand the information provided to them, especially in an environment that lacks direct, face-to-face interaction with healthcare providers. Gazmararian et al. [25] argue that it is both unethical and unprofessional for healthcare providers to deliver complex and technical information in ways that are accessible to an average and ideal participant [26]. This issue is discussed by Rowlands and Nutbeam [27] as the “inverse information law,” where those with the lowest levels of health literacy have the least access to comprehensible health information [28]. For instance, digital platforms may not be designed to accommodate individuals with low literacy, cognitive impairments, or language barriers. The increasing reliance on digital health solutions thus creates an ethical imperative to bridge the gap in health literacy. Remote systems must be designed to be user-friendly and inclusive, providing concise information in multiple formats (text, video, and audio), to meet different interlocutors [29]. Another key aspect is the role of digital literacy, which overlaps with health literacy. Even if participants are health-literate, they may struggle with navigating the technological aspects of the consent process, further complicating their ability to provide informed consent. Addressing the issue of health literacy, particularly in remote informed consent, is not just a technical challenge, it is a matter of equity and continuous updates are needed to promote they remain accessible as technology and participant needs evolve [30]. Health providers and clinical trial organizers must take initiative-taking steps to create systems that support informed, ethical decision-making and address the disparities that may arise.

While technical solutions such as tutorials and digital platforms can support comprehension, they do not fully resolve the ethical issues at stake. Ensuring equitable access and inclusion is essential: all participants, regardless of health literacy, digital literacy, cognitive abilities, or language proficiency, must have the opportunity to understand and meaningfully engage with the consent process. The ethical imperative is to create systems that not only facilitate information delivery, but also actively promote fairness, comprehension, and respect for participants’ autonomy in decentralized clinical trials.

**Table 1 medsci-13-00222-t001:** The central role of health literacy in the successful implementation of DCTs, with a particular focus on cardiovascular research.

Aspect	Description/Challenge	Digital Solutions in DCTs	Impact on Cardiovascular Participants
Understanding information [25,27]	Participants with low health literacy struggle to understand study risks, benefits, and procedures	Interactive platforms with simplified text, explanatory videos, charts, and diagrams	Better trial adherence, reduced anxiety, and increased active participation
Digital literacy [29]	Difficulty using apps, wearables, and digital platforms	Guided tutorials, real-time support, multilingual assistance	Improved remote monitoring of blood pressure, heart rhythm, and other parameters
Autonomous engagement	Informed decision-making hindered by complex information	Interactive e-Consent with identity verification and consent traceability	Cardiovascular participants can clearly understand intervention or medication risks, increasing safety and trust
Accessibility [31]	Language, cognitive, or geographic barriers	Translations, subtitles, cultural adaptations, remote assistance	Inclusion of underrepresented groups, expanding diversity in cardiovascular trials
Continuous monitoring [32]	Limited understanding of self-management procedures	Wearables and telemonitoring with automatic alerts and educational feedback	Better control of critical parameters such as blood pressure, heart rhythm, and physical activity

### 2.6. Medico-Legal Considerations

The new methods of conducting DCTs pose multidimensional challenges, particularly medico-legal, where the digital transformation of traditional approaches necessitates a radical rethinking of normative and accountability paradigms [33]. However, in an increasingly supranational perspective, the regulatory perimeter in the health sector of the same changed activities must think of new instruments or updates [34,35]. The need to guarantee the identity and legitimacy of each participant, along with the ability to indisputably document every phase of the process, suggests the adoption of emerging technologies such as blockchain, which can serve as a tool for accountability by providing an immutable and verifiable structure for managing data and digital consents, without being the primary focus of the trial itself [36].

The use of digital platforms for obtaining informed consent, remotely monitoring participants, and employing advanced algorithms for the analysis of clinical data, opens a different and new role for healthcare professionals and other stakeholders involved in clinical research. Digitalization is in fact redefining the traditional boundaries of civil and criminal liability [37]. Simultaneously, the transnational nature of decentralized trials introduces further challenges in terms of jurisdictional competence and regulatory harmonization. Then, the increasing interconnection between clinical, technological, and legal aspects demands an integrated approach to the governance of clinical trials, in which responsibility is not limited to the formal fulfilment of regulatory requirements but extends to the adoption of innovative strategies aimed at protecting participants’ rights and safeguarding their dignity. This implies that the entities involved in managing studies must invest in advanced cybersecurity systems, specialized training programs for healthcare and legal professionals, and procedures that enable comprehensive and transparent documentation of every activity—from the collection of consent to the sharing of clinical research data. This paradigm of “privacy by design” and “security by design” becomes crucial for preventing breaches and ensuring that every technological innovation translates into a real benefit for the participant, thereby minimizing the risks of errors, manipulations, or non-compliance. Finally, while the evolution of digital technologies offers the possibility to expand the scope and effectiveness of clinical trials, it also requires constant revision of operational practices and regulations so that these can anticipate emerging risks and adapt to the rapid changes in the global healthcare context [32]. Ensuring privacy and security in decentralized trials goes beyond mere regulatory compliance. Protecting participants’ data and digital consent is fundamentally a matter of safeguarding their rights, autonomy, and dignity. Any breach, error, or misuse of data carries moral implications, as it can undermine trust, cause harm, or violate the ethical commitment researchers have toward participants. Therefore, the adoption of robust cybersecurity measures, transparent documentation, and secure technological solutions should be understood not only as procedural obligations but as essential ethical responsibilities that uphold the integrity of the research and protect participants from potential harm.

### 2.7. Specific Challenges to Cardiovascular Trials

Cardiovascular trials often involve high-risk participants, with several comorbidities and high-complexity procedures, which require highly specialized monitoring and care. Ensuring that participants are effectively managed in a decentralized setting might require innovative solutions, such as advanced telemedicine platforms or specialized remote healthcare teams [38]. Furthermore, the implementation of decentralized trials has the potential to address the necessity for postoperative care and follow-ups, as well as the requirement for longitudinal, long-term studies to evaluate the efficacy of surgical techniques and drugs. Wearables, advanced telemedicine tools, AI-assisted monitoring, and home healthcare services will continue to enhance the feasibility of DCTs in this field, but generating large amount of data [31,39]. Continuous cardiac conduction or blood pressure monitoring are some examples. The a priori DCT data digitization could meet not only the needs of the trial, but also the need for reliable, accurate and comprehensive data to evaluate outcomes, especially where specific interventions or rare diseases are concerned, with implications for participant safety and health policy. Decentralized approaches have the potential to recruit and retain previously excluded or underrepresented groups in cardiovascular clinical research, avoiding the potential for study populations that lack diversity in race, ethnicity, gender, age, geography, comorbidities, and socioeconomic status to exacerbate health inequalities [40]. In addition, they can reduce disproportionate risks and health inequalities within populations and countries, although territorial characteristics also need to be considered by health actors (Table 2). Participants in cardiovascular trials are particularly vulnerable due to their health status and complex care needs. There is a potential risk of implicit coercion, as participants may feel pressured to consent or continue participation because of perceived authority or dependence on healthcare providers. This risk is heightened when clinicians serve a dual role as both caregivers and researchers, which may influence participants’ decisions. Ethical trial design must therefore include safeguards to protect autonomy, ensure clear and understandable communication, and actively minimize the risk of undue influence, while maintaining rigorous monitoring and support tailored to participants’ vulnerabilities.

## 3. Conclusions

DCT now represents an important opportunity for research. Digital tools have allowed us to solve the issues of increasing numerosity, representativeness, and validity of clinical trials, with some issues to be acknowledged and managed in the building phase of the trial. European data space with the health ones must be managed with GDPR, devices, and overall health regulations compliance. Informed consent for health research is the cornerstone of this process, and participants in DCTs must be made aware of the research purpose and the specific features of the process, which should be properly disclosed. The informed consent phase is the most important action in healthcare, making the patient a key actor, aware of and responsible for the entire healthcare process. DCTs are particularly important for the cardiovascular population, ensuring participation, long-term outcomes monitoring and sampling bias reduction, thereby improving the quality of care. It should be noted that centralization is not necessarily the gold standard; decentralized clinical trials can, in many cases, offer a more ethical and scientifically robust approach, by enhancing participant engagement, improving representativeness, and reducing barriers to inclusion. Beyond technical management, DCTs also demand reflection on ethical dimensions: promoting genuine participant autonomy, ensuring equitable access and inclusion, fostering meaningful communication, and protecting participants’ rights and dignity throughout the research process. This perspective emphasizes that DCTs are not only a logistical or technological solution, but also an opportunity for ethically strengthened and participant-centered clinical research.

## Figures and Tables

**Figure 1 medsci-13-00222-f001:**
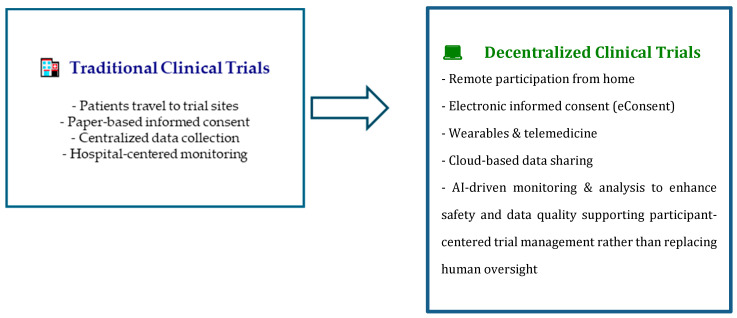
Evolution of Clinical Trials: From Traditional to Decentralized.

**Figure 2 medsci-13-00222-f002:**
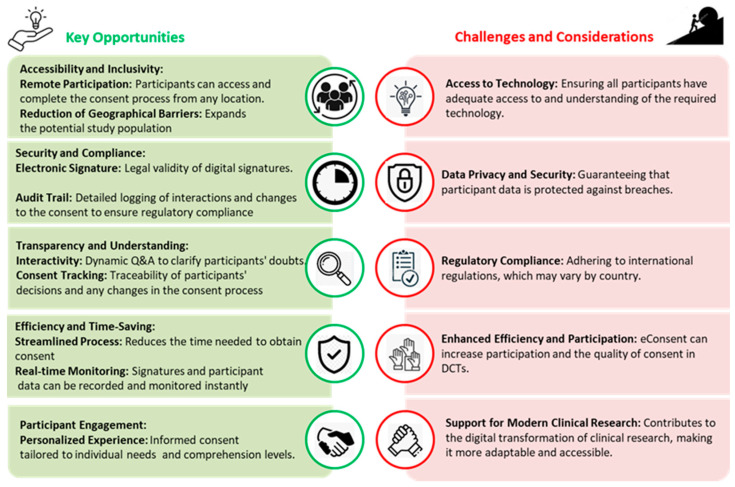
Electronic informed consent for DCT.

**Table 2 medsci-13-00222-t002:** Ethical, Legal and Organizational Challenges of Decentralized Clinical Trials (DCTs) in the Cardiovascular Field.

Domain	Challenge	Proposed Solution/Best Practice
Ethical and medico-legal	Adequate understanding of informed consent in digital settings	e-Consent with simplified language, videos/infographics, multilingual support, interactive Q&A
Legal	GDPR compliance and clear identification of data controller and data processor	Privacy by design, DPO appointment, access privilege definition, audit trail
Technological	Data security and traceability	Blockchain, advanced cryptography, immutable logs
Organizational	Transnational coordination and regulatory heterogeneity	Harmonization through EMA/ACT EU guidance, shared procedures
Clinical	Monitoring of fragile participants (e.g., complex cardiovascular cases)	Advanced telemedicine, wearable devices, remote healthcare teams
Equity	Inclusion of vulnerable populations with low health or digital literacy	User-friendly platforms, linguistic/cultural support, basic digital training

## Data Availability

No new data were created or analyzed in this study.

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
