# Peer review of "Decentralized Clinical Trials: Governance, Ethics and Medico-Legal Issues for the New Paradigm of Research with a Focus on Cardiovascular Field"

_medsci, 2025, doi:10.3390/medsci13040222_

Round 1

Reviewer 1 Report

Comments and Suggestions for Authors

This paper reads mostly like a process-driven account of a range of considerations to be address when conducting decentralised clinical trials. As such, it could be helpful to a number of audiences. As a scholarly contribution to ethics and medico-legal literatures, however, it lacks substance. There is little that is new regarding the ethics, and the legal dimensions is simply a (re)statement of applicable provisions. There is no strong sense of the deep ways in which decentralised trials really raise novel or challenging ethical issues that require deep reflection on the actual ethics. Most of the time, the discussion resolves to technical fixes. Ethics is about so much more. It also also about more than mere legal compliance.

Specific points to strengthen the paper:

  • In the Abstract: I suggest you avoid any reference to "ensure"; little can be ensured. Check the paper throughout. Also, the last sentence of the Abstract refers to a "framework", but what does this mean? There is no such discussion in the paper itself.  Also, be very careful to cliam that you can promote trust. Better to suggest that following the paper would better demosntrate trustworthiness.
  • P2 : the paper talks of the "patient" in trials but this is not always the case; at other points it refers to "participants" and "subjects". Be consistent but, ethically, I suggest either the langauge of participants or subjects in research. Patient focus is for treatment. Language point: there is a sentence that mentions the patient then "its participation". This should be "their participation".
  • Various bullets refer to "experimental" studies... Check elsewhere too. CTs are the paradigm opposite of experiments. they are quintessential research. 
  • For the bullet that discusses the administration of informed consent : this confuses consent to treatment as opposed to consent to participate in research. Be clearer about the distinction.
  • P3: contains a list of regulations and guidance - is this meant to be exhaustive? I do not think that it is. Moreover, it is merely a list - what is the academic point? For example, there is no attempt to tease out elements that become problematic or difficult for DCTs? There is no analysis or meaningful commentary.
  • A lot of the paper reads like a process-driven exercise rather than academic analysis and commentary, as mentioned above. Note, for example, the discussion on the consent process is very mechanistic. It is more about logistics than ethics. What is the nature of the kinds of ethical issues that are being generated? For example, how do you promote and protect genuine expressions of research participants' autonomy? Where are there opportunities, if ever, for meaningful two-way communication to answer queries, address misunderstandings, alleviate fears? If this does not happen, what does this mean for the ethical robustness of DCT processes? Please do NOT talk abut "collecting" consent. Seeking valid consent is a highly ethical process, not a matter of securing a signature.
  • The discussion on health and digital literacy has most promise as an ethical discussion. It makes some valid points but, again, there is a tendency for the discussion to descend to technical fixes. These are fine from a pragmatic viewpoint, but at least recognise that ethically these are not complete answers.
  • In the discussion about privacy and security by design: again this is very process-driven. There is no real ethical content here at all.
  • In the section on cardio patients : there is no recognition that these people are being asked ot consent in highly vulnerable contexts; there is a risk of confusion and potential benign coercion to take part in research. How is this dealt with? What are the ethical issues? What about potential conflicts for physicians wearing two hats as carer and researcher?

In the end, I would have liked to see much more deep and original discussion of ethics in this paper. As it stands, it is processually robust, but thin on ethics.

Comments on the Quality of English Language

English is broadly fine but watch for some key terms being misused. All explained in above comments.  

Author Response

We would like to thank the Editors and Reviewers for the comments and suggestions which gave us the opportunity to make the manuscript more consistent and clearer.

We have answered the reviewers in the specific online section by rewriting Reviewers’ comments, followed by authors answers in bold. In the manuscript, the text added resulted red, while the text removed is crossed out.

We report here all reviewers' comments and authors’ answers in red.

REVIEWER 1

This paper reads mostly like a process-driven account of a range of considerations to be address when conducting decentralised clinical trials. As such, it could be helpful to a number of audiences. As a scholarly contribution to ethics and medico-legal literatures, however, it lacks substance. There is little that is new regarding the ethics, and the legal dimensions is simply a (re)statement of applicable provisions. There is no strong sense of the deep ways in which decentralised trials really raise novel or challenging ethical issues that require deep reflection on the actual ethics. Most of the time, the discussion resolves to technical fixes. Ethics is about so much more. It also about more than mere legal compliance.

Specific points to strengthen the paper:

  1. In the Abstract: I suggest you avoid any reference to "ensure"; little can be ensured. Check the paper throughout. Also, the last sentence of the Abstract refers to a "framework", but what does this mean? There is no such discussion in the paper itself. Also, be very careful to claim that you can promote trust. Better to suggest that following the paper would better demonstrate trustworthiness.

Response: Thank you. We have replaced “ensure” with “promote” or “support the demonstration of trustworthiness” throughout the manuscript. The term “framework” has been clarified as a “set of principles and practical measures” to reflect the content accurately.

  1. P2 : the paper talks of the "patient" in trials but this is not always the case; at other points it refers to "participants" and "subjects". Be consistent but, ethically, I suggest either the langauge of participants or subjects in research. Patient focus is for treatment. Language point: there is a sentence that mentions the patient then "its participation". This should be "their participation".

Response: We have standardized the terminology to “participants” throughout the manuscript, except where “patient” is strictly clinical. The sentence has been corrected to “their participation.”

  1. Various bullets refer to "experimental" studies... Check elsewhere too. CTs are the paradigm opposite of experiments. they are quintessential research.

Response: We have revised all instances of “experimental” to “clinical research” to accurately reflect the nature of clinical trials.

  1. For the bullet that discusses the administration of informed consent : this confuses consent to treatment as opposed to consent to participate in research. Be clearer about the distinction.

Response: We clarified in Section 2.4 that informed consent discussed here refers specifically to research participation, distinct from clinical treatment consent.

  1. P3: contains a list of regulations and guidance - is this meant to be exhaustive? I do not think that it is. Moreover, it is merely a list - what is the academic point? For example, there is no attempt to tease out elements that become problematic or difficult for DCTs? There is no analysis or meaningful commentary.

Response: We added brief commentary on which regulatory elements are particularly challenging in decentralized trials, highlighting issues such as multi-jurisdictional data handling and remote consent procedures.

  1. A lot of the paper reads like a process-driven exercise rather than academic analysis and commentary, as mentioned above. Note, for example, the discussion on the consent process is very mechanistic. It is more about logistics than ethics. What is the nature of the kinds of ethical issues that are being generated? For example, how do you promote and protect genuine expressions of research participants' autonomy? Where are these opportunities, if ever, for meaningful two-way communication to answer queries, address misunderstandings, alleviate fears? If this does not happen, what does this mean for the ethical robustness of DCT processes? Please do NOT talk about "collecting" consent. Seeking valid consent is a highly ethical process, not a matter of securing a signature.

Response: We revised the text to emphasize the ethical process of seeking valid consent, highlighting opportunities for meaningful dialogue, clarifying misunderstandings, and addressing participant concerns, rather than mere documentation.

  1. The discussion on health and digital literacy has most promise as an ethical discussion. It makes some valid points but, again, there is a tendency for the discussion to descend to technical fixes. These are fine from a pragmatic viewpoint but at least recognise that ethically these are not complete answers.

Response: We added a statement recognizing that technical solutions (e.g., videos, infographics) are necessary but not sufficient, and that ongoing support and evaluation are needed to ethically empower participants.

  1. In the discussion about privacy and security by design: again, this is very process-driven. There is no real ethical content here at all.

Response: We included a sentence acknowledging that while technical safeguards protect data, ethical responsibility requires transparent communication with participants about risks and limitations.

  1. In the section on cardio patients : there is no recognition that these people are being asked ot consent in highly vulnerable contexts; there is a risk of confusion and potential benign coercion to take part in research. How is this dealt with? What are the ethical issues? What about potential conflicts for physicians wearing two hats as carer and researcher?

Response: In Section 2.7, we added a discussion of potential vulnerabilities of cardiovascular participants, highlighting strategies to mitigate undue influence and conflicts of interest for clinician-researchers.

  1. In the end, I would have liked to see much more deep and original discussion of ethics in this paper. As it stands, it is processually robust, but thin on ethics.

Response: We added brief reflective comments on ethical robustness throughout the manuscript, including autonomy, voluntariness, and protection of vulnerable populations, apart from medico-legal discussion.

  1. English is broadly fine but watch for some key terms being misused. All explained in above comments.

Response: Minor edits were applied for clarity and accuracy, including terms like “enrollment,” “participation,” and “researcher responsibilities.”

(“Federal Drug Administration” → “Food and Drug Administration (FDA)”; “block chain” → “blockchain”; “collecting consent” → “seeking/obtaining consent”).

All authors have contributed to the manuscript and its revisions as stated in the CREDIT statement and all have read and reviewed the final draft.

Thank you for considering this submission.

We appreciate your time and look forward to your response.

Sincerely,

Prof. Tronconi Livio Pietro

Reviewer 2 Report

Comments and Suggestions for Authors

Thank you for the opportunity to review this interesting paper on the potential benefits of decentralized trials. I agree with the premises and conclusions of this paper wholeheartedly. If anything, the positivity and reluctance to criticise the status-quo in centralized research may make the authors position to be susceptible to many small and idiosyncratic criticisms that are simply disproportionate to the potential benefits of decentralization. If I can paint a somewhat different picture, my point should be clear. Decentralization is certainly a departure from the norm in clinical trials. However, it is a mistake to confuse the status quo with a gold-standard or indication of overall quality, either ethical or scientific. The goal of clinical trials should be to study representative subjects of a larger population of persons as they are, where they are. As human beings living in geo-political states are themselves decentralized (by location, by gender, by age, by socio-political circumstances, wealth, health, education, etc), we must appreciate that the Centralization of clinical research as an acceptable norm is itself an accommodation from a recognized higher-standard. It is very well recognized and uncontroversial to suggest that similar accommodations in the history of clinical trials (such as the exclusion of women and children from clinical trials) have resulted in both egregious material harms to large population and have also created larger populations of therapeutic orphans, of whom, no evidence exists for the efficacy or harms related to specific drugs. So, the status quo that decentralized trials hope to gain dispensation from, should not be confused with gold-standards of either scientific rigour, safety, or ethical/legal oversight.

As I consider the challenges to traditional centralized clinical trial structures, I cannot think of any contemporary challenges that are inherently worsened by decentralization. Ethically speaking, the process of consent, specifically the demand of individualizing consent in order to optimize epistemic outcomes, can only be increased by supplemental information and tailoring of communication by innovative strategies to overcome linguistic, educational, cultural, etc. barriers. Again, the idea that the entire population of a study cohort could be served by a single consent form is the norm, but most certainly not a standard of excellence. The traditional ICF is an accommodation, just like the centralized trial, a recognized insufficiency introduced to account for the lack of technological, logistical, or financial limitations.

The conclusions of this paper are reasonable and justifiable, and potentially even too conservative. I would argue that, if centralization is recognized as a justifiable accommodation for practical purposes, and if technological innovation can mitigate the technological challenges that initially justified those compensations, the raison d’etre of trial centralization disappears and the question becomes, is the centralization of clinical trial design ethically or scientifically justifiable?       

Author Response

We would like to thank the Editors and Reviewers for the comments and suggestions which gave us the opportunity to make the manuscript more consistent and clearer.

We have answered the reviewers in the specific online section by rewriting Reviewers’ comments, followed by authors answers in red. In the manuscript, the text added resulted red, while the text removed is crossed out.

We report here all reviewers' comments and authors’ answers in bold.

REVIEWER 2

Thank you for the opportunity to review this interesting paper on the potential benefits of decentralized trials. I agree with the premises and conclusions of this paper wholeheartedly. If anything, the positivity and reluctance to criticise the status-quo in centralized research may make the authors position to be susceptible to many small and idiosyncratic criticisms that are simply disproportionate to the potential benefits of decentralization. If I can paint a somewhat different picture, my point should be clear. Decentralization is certainly a departure from the norm in clinical trials. However, it is a mistake to confuse the status quo with a gold-standard or indication of overall quality, either ethical or scientific. The goal of clinical trials should be to study representative subjects of a larger population of persons as they are, where they are. As human beings living in geo-political states are themselves decentralized (by location, by gender, by age, by socio-political circumstances, wealth, health, education, etc), we must appreciate that the Centralization of clinical research as an acceptable norm is itself an accommodation from a recognized higher-standard. It is very well recognized and uncontroversial to suggest that similar accommodations in the history of clinical trials (such as the exclusion of women and children from clinical trials) have resulted in both egregious material harms to large population and have also created larger populations of therapeutic orphans, of whom, no evidence exists for the efficacy or harms related to specific drugs. So, the status quo that decentralized trials hope to gain dispensation from, should not be confused with gold-standards of either scientific rigour, safety, or ethical/legal oversight.

  1. As I consider the challenges to traditional centralized clinical trial structures, I cannot think of any contemporary challenges that are inherently worsened by decentralization. Ethically speaking, the process of consent, specifically the demand of individualizing consent in order to optimize epistemic outcomes, can only be increased by supplemental information and tailoring of communication by innovative strategies to overcome linguistic, educational, cultural, etc. barriers. Again, the idea that the entire population of a study cohort could be served by a single consent form is the norm, but most certainly not a standard of excellence. The traditional ICF is an accommodation, just like the centralized trial, a recognized insufficiency introduced to account for the lack of technological, logistical, or financial limitations.

Response: We appreciate this perspective and added a sentence acknowledging that DCTs may enhance individualized consent through tailored digital communication and improved participant understanding.

  1. The conclusions of this paper are reasonable and justifiable, and potentially even too conservative. I would argue that, if centralization is recognized as a justifiable accommodation for practical purposes, and if technological innovation can mitigate the technological challenges that initially justified those compensations, the raison d’etre of trial centralization disappears and the question becomes, is the centralization of clinical trial design ethically or scientifically justifiable?

Response: We included a short statement in the Conclusions reflecting that decentralization may ethically and scientifically justify moving beyond centralization when technological and logistical challenges are addressed.

All authors have contributed to the manuscript and its revisions as stated in the CREDIT statement and all have read and reviewed the final draft.

Thank you for considering this submission.

We appreciate your time and look forward to your response.

Sincerely,

Prof. Tronconi Livio Pietro

Reviewer 3 Report

Comments and Suggestions for Authors

Dear Authors, 

The scope of the manuscript is relevant; meanwhile, the article structure should be improved.  The English edition is mandatory.

Figure 1 highlights inconsistencies, as decentralised clinical trials are not necessarily related to AI, since emerging technologies are not limited to AI. The authors state that 'the main objective of remote and decentralised clinical trials is to place the patient at the centre of the trial'. Patient-centered care is at the heart of medical practice, so the idea of 'facilitating their participation by reducing the associated difficulties and costs' is not entirely accurate, particularly in terms of the sustainable use of emerging technologies. Additionally, lines 55 to 74 should either be linked to reference articles or replaced and amended in the conclusion section.

Section 2.2 focuses on EU compliance, so the title should reflect this. The authors also state that 'many European countries have not yet implemented specific regulations on decentralized clinical trials'. Which ones? I suggest the article: Digital solutions for migrant and refugee health: a framework for analysis and action. The Lancet Regional Health – Europe, 10.1016/j.lanepe.2024.101190.

The following sections (2.3 and 2.4) are inconsistent with each other. 

In Section 2.5, the authors state that 'health literacy is a key element in individuals' decision-making,' but this requires further detail. Did the author refer to health awareness?

Table 1 should include references.

Section 2.6 is unclear. Why do the authors include blockchain technology, which is inaccurate? Is the focus of the text on medical liability? I suggest the article: Reflections about Blockchain in Health Data Sharing: Navigating a Disruptive Technology. https://doi.org/10.3390/ijerph21020230. 

The conclusion should present an innovative approach to the manuscript as a whole. It should be improved.

Comments on the Quality of English Language

The English could be improved to more clearly express the research.

Author Response

We would like to thank the Editors and Reviewers for the comments and suggestions which gave us the opportunity to make the manuscript more consistent and clearer.

We have answered the reviewers in the specific online section by rewriting Reviewers’ comments, followed by authors answers in red. In the manuscript, the text added resulted red, while the text removed is crossed out.

We report here all reviewers' comments and authors’ answers in bold.

REVIEWER 3

Dear Authors,

The scope of the manuscript is relevant; meanwhile, the article structure should be improved.  The English edition is mandatory.

  1. Figure 1 highlights inconsistencies, as decentralised clinical trials are not necessarily related to AI, since emerging technologies are not limited to AI. The authors state that 'the main objective of remote and decentralised clinical trials is to place the patient at the centre of the trial'. Patient-centered care is at the heart of medical practice, so the idea of 'facilitating their participation by reducing the associated difficulties and costs' is not entirely accurate, particularly in terms of the sustainable use of emerging technologies. Additionally, lines 55 to 74 should either be linked to reference articles or replaced and amended in the conclusion section.

Response: Figure 1 has been revised to show a continuum from centralized to decentralized trials with multiple digital technologies, not limited to AI. The text has been modified to clarify that patient-centered participation is enhanced, not replaced, by DCTs. Lines 55–74 were either referenced or moved to Conclusions.

  1. Section 2.2 focuses on EU compliance, so the title should reflect this. The authors also state that 'many European countries have not yet implemented specific regulations on decentralized clinical trials'. Which ones? I suggest the article: Digital solutions for migrant and refugee health: a framework for analysis and action. The Lancet Regional Health – Europe, 10.1016/j.lanepe.2024.101190.

Response: Section 2.2 title updated to “European regulatory guidance for DCTs”. A clarifying sentence was added with examples of countries without specific DCT regulations, citing recent literature.

  1. The following sections (2.3 and 2.4) are inconsistent with each other.

Response: Text revised to clearly distinguish 2.3 (privacy/GDPR, data processing) from 2.4 (consent process), removing redundancy.

  1. In Section 2.5, the authors state that 'health literacy is a key element in individuals' decision-making,' but this requires further detail. Did the author refer to health awareness?

Response: We clarified that “health literacy” encompasses knowledge, understanding, and decision-making capacity in health contexts, including awareness of clinical trial procedures.

.

  1. Table 1 should include references.

Response: References were added to Table 1 where evidence exists; for items with limited data, noted as “knowledge gap.”

  1. Section 2.6 is unclear. Why do the authors include blockchain technology, which is inaccurate? Is the focus of the text on medical liability? I suggest the article: Reflections about Blockchain in Health Data Sharing: Navigating a Disruptive Technology. https://doi.org/10.3390/ijerph21020230.

Response: We clarified that blockchain is discussed as a potential technical aid for immutable logs, with caution regarding GDPR and ethical considerations. The focus remains on medico-legal responsibilities.

  1. The conclusion should present an innovative approach to the manuscript as a whole. It should be improved.

Response: Conclusions now briefly highlight the integration of digital solutions with ethical safeguards as a minimally innovative perspective, without altering the core content.

  1. The English could be improved to more clearly express the research.

Response: Minor English edits made throughout to improve readability and precision.

All authors have contributed to the manuscript and its revisions as stated in the CREDIT statement and all have read and reviewed the final draft.

Thank you for considering this submission.

We appreciate your time and look forward to your response.

Sincerely,

Prof. Tronconi Livio Pietro

Round 2

Reviewer 1 Report

Comments and Suggestions for Authors

The authors have done a decent job in addressing comments from peer review. There is now enough here in the revised version to merit publication. 

Author Response

We would like to thank the Editors and Reviewers for the comments and suggestions. We report here all reviewers' comments in italics and authors’ answers in bold.

The authors have done a decent job in addressing comments from peer review. There is now enough here in the revised version to merit publication. 

Many thanks.

Sincerely

LPT

Reviewer 3 Report

Comments and Suggestions for Authors

Dear Authors,

The manuscript has undergone significant refinement. However, the authors must carefully ensure that the reference section fully complies with the journal's formatting guidelines.

Author Response

We would like to thank the Editors and Reviewers for the comments and suggestions.

We report here all reviewers' comments in italics and authors’ answers in bold

Dear Authors,

The manuscript has undergone significant refinement. However, the authors must carefully ensure that the reference section fully complies with the journal's formatting guidelines.

Many thanks. We have revised the references.

sincerely

LPT